# Multilayer Packaging in a Circular Economy

**DOI:** 10.3390/polym14091825

**Published:** 2022-04-29

**Authors:** Jannick Schmidt, Laura Grau, Maximilian Auer, Roman Maletz, Jörg Woidasky

**Affiliations:** 1Institut für Industrial Ecology, Hochschule Pforzheim, Tiefenbronner Straße 65, 75175 Pforzheim, Germany; maximilian.auer@hs-pforzheim.de (M.A.); joerg.woidasky@hs-pforzheim.de (J.W.); 2Institute of Waste Management and Circular Economy, Technische Universität Dresden, Pratzschwitzer Straße 15, 01796 Pirna, Germany; roman.maletz@tu-dresden.de

**Keywords:** multilayer, lightweight packaging, circular economy, plastic, recycling

## Abstract

Sorting multilayer packaging is still a major challenge in the recycling of post-consumer plastic waste. In a 2019 Germany-wide field study with 248 participants, lightweight packaging (LWP) was randomly selected and analyzed by infrared spectrometry to identify multilayer packaging in the LWP stream. Further investigations of the multilayer packaging using infrared spectrometry and microscopy were able to determine specific multilayer characteristics such as typical layer numbers, average layer thicknesses, the polymers of the outer and inner layers, and typical multilayer structures for specific packaged goods. This dataset shows that multilayer packaging is mainly selected according to the task to be fulfilled, with practically no concern for its end-of-life recycling properties. The speed of innovation in recycling processes does not keep up with packaging material innovations.

## 1. Introduction

For years, the quantities of plastic waste in Germany have been steadily increasing. In 1994, plastic waste generation was 2.8 million t [1], but by 2019 it had already risen to 6.28 million t [2]. The average annual growth of about 3.3% is almost exclusively due to the waste generated in the post-consumer sector, with the packaging industry being by far the largest consumer of plastics in Germany [2]. Consequently, more than 50% of the plastic waste generated today in Germany can be attributed to short-lived packaging [3]. Plastic packaging waste almost doubled from 1991 (1.64 million t) to 2017 (3.18 million t), even though individual plastic packaging items, in general, became on average 25% lighter during this period [4,5]. Accordingly, plastic packaging is perceived by the public as one of the biggest environmental problems [6], and has thus led to the adoption of stricter environmental laws to reduce plastic packaging and increase the recycling rate [7]. In Germany, the Packaging Act (VerpackG), which came into force in 2019, stipulates that mechanical recycling rates must be 63% by 2022 [8].

To strengthen a more sustainable approach to plastics, the EU presented “A European Strategy for Plastics in a Circular Economy (CE)” in 2018 [9]. This stipulates that all plastic packaging placed on the market in the EU should be either reusable or recyclable in a cost-effective manner by 2030 [9]. This approach is intended to break the current prevailing linear flow (open loop) of plastics along the value chain from production to use and disposal [10], as this is one of the main sources of CO_2_ emissions and pollution [11]. It is estimated that 95% of the value of plastic packaging is lost after the first phase of use [12]. This is due to the use of mechanical recycling to reprocess mixed plastic waste streams, which leads to a decrease in molecular mass and thus limits the number of possible reprocessing processes [13]. Recycled plastics can therefore often only be downcycled [14] and are primarily used to produce products other than those originally made from the material [15]. Such intensive use of finite resources for a linear economic model of production, use, and disposal is proving to be unsustainable [16]. The implementation of a true CE (closed loop) in the field of plastics could help in reducing downcycling, incineration, and landfill, allowing plastic waste to be recycled back into the same or equivalent new products [11]. However, before a true CE based on a balance between economic, environmental, and social impacts can be achieved [17], the prevailing problems must be overcome. Well-functioning plastic recycling processes are needed, in order to move from a linear flow to a closed loop [18].

Along with the requirements for suitable CE packaging, the requirements for food packaging are also steadily increasing. The basic packaging requirements relate to strength and sealability, machinability (softening, slip, rigidity, pliability, and heat resistance), promotion, and convenience [19], as well as barrier properties against oxygen, water vapor, light, carbon dioxide, and flavoring substances, which enable a long shelf life and thus the current form of food trade and reduced food losses [20,21,22]. For this purpose, multilayer packaging is often used. Multilayer packaging combines different polymeric and non-polymeric materials such as paper or aluminum [23,24,25], which enables customized property profiles with low material consumption [26]. Multilayers can reduce the cost of existing film structures, e.g., by replacing expensive polymers with less costly ones, reducing film thickness, or using recycled materials [27]. Furthermore, the combination of different layers achieves a functionality that is not possible through the use of a single layer [28]. According to estimates, about 17% [23] to 20% [29] of plastic packaging consists of multilayer packaging, and it is increasingly used in the packaging of food, pharmaceuticals, medicines, cosmetics, and electronics [24,30].

While multilayer packaging does not differ from mono-material solutions in terms of use and collection, challenges become obvious in packaging sorting plants, as multilayer packaging is difficult to identify and hard to recycle [31]. Here, spectroscopic identification technology (NIR) can only identify surface properties, and thus, for physical reasons, does not identify multilayers properly. In subsequent recycling processes, recovery of multilayer material becomes possible if either the materials can be separated or they can be processed jointly. However, this is not easy to achieve because, on the one hand current recycling systems are aimed at recycling mono-materials, and on the other hand the different polymers or materials are often immiscible. Therefore, multilayer packaging is considered non-recyclable, and only thermal recovery or final disposal routes remain [14,19,32,33].

The wide variety of achievable properties through the combination of different numbers of layers, layer materials, and layer thicknesses cannot be properly managed with current waste management technologies and systems. Information regarding the individual material composition of packaging is required to improve current identification and recycling technologies. While this information is available from the producers, it is currently lost along the supply chain. Standardized recycling codes do not provide a sufficiently detailed level of information, and other indications provided by the manufacturer are rarely disclosed on the packaging media, as the composition often represents an important competitive advantage. Consequently, polymer multilayers are marked with the recycling code “Other” (07) according to DIN 6120 (German Institute for Standardization) (After the revision of DIN 6120, the additional designation 07 (“Other”) was deleted.) or not marked at all [34]. This is insufficient for the future development of waste management systems and substantial material recycling quota increases, as technology and strategy development rely on large and precise amounts of data [35]. Currently, the lack of data leads to barriers in choosing the most appropriate strategy to close material and product loops [36]. Great potential is therefore seen in the area of big data applications, where comprehensive data-driven decision-making can also take into account the integration of sustainability approaches across supply chain networks, to pave the way toward a CE [37,38].

Waste characterization is key to reprocessing high-value end products [39]. This has become particularly clear since digitization has entered the waste management sector and knowledge in the form of data on the type and composition of individual LWP (lightweight packaging) wastes has come into focus. The combination with digital technologies enables companies to improve the circularity of their systems [40]. As early as 2017, a survey of 394 companies from the waste management and recycling technology sectors in Germany, Austria, and Switzerland revealed that 63% of those surveyed perceived digitization in the company as an opportunity for further development [41], and this was despite the fact that actual digital readiness, which describes the degree of digital transformation of companies in the waste management sector in German-speaking countries, is only at 30% [42]. This is reflected, for example, in the fact that many plant operators operate their sorting facilities with fixed parameter settings based on empirical values, without any physical or statistical proof of optimality, which is ultimately due to the complexity involved in the study of mixed solid waste [43]. There are already projects and companies working on developing new technological approaches to addressing the complexity of digitally assisted optimization of LWP waste sorting in an industrial setting. These approaches make an important contribution to the transition of the current linear system to a CE by capturing valuable resources that would normally be lost under the current linearity [44].

With this in mind, and motivated by the need for waste management to shift from a linear to a circular approach [45], the research question in this work concerns how multilayer LWP items, forming a substantial fraction of post-consumer packaging waste, can be characterized in detail. Consequently, the scope of this work is to collect, analyze, and publish key data on multilayer packaging applications as a fraction (class of items) of plastic LWP waste, as a basis for future waste management system developments and recycling rate increases. A more holistic view of the treatment of post-consumer LWP waste has been published in [46]. With these results, recommendations to the production industry can be made to increase the recyclability and circularity of this material stream.

## 2. Multilayer Packaging: State of the Art

### 2.1. Structure of Multilayer Packaging

Multilayer packaging can be flexible or semi-rigid and involve polymeric layers as well as inorganic layers such as Al or SiOx coatings. It usually consists of from 2 [47] up to 24 layers [48]. Each layer adds an important function to the overall architecture. The intended functions and the layers used to achieve them, as well as the materials used to fulfill these functions, are listed in Table 1.

The seal layer (1) is in direct contact with the packaged goods, therefore good migration limitation and an interaction barrier must be provided, for example by using a polyolefin-type inner layer [28]. For freshness conservation, the permeability of the packaging material against specific substances, or generally to prevent any gaseous and liquid exchange, often needs to be adjusted.

This can be enabled by increasing the layer thickness or just by using suitable *barrier layers* (2) in between to prevent oxidation, microbial spoilage, loss or gain of moisture, and both loss of flavor and aroma or gain of unwanted aromas from outside [26]. *The oxygen barrier* avoids the packaged goods becoming rancid, as well as the proliferation of aerobic microorganisms. A *light*-*barrier layer* such as an aluminum coating or a TiO_2_-filled polymer layer can also help to conserve freshness [19,24,26]. *Migration barriers* may be of the utmost importance in multilayer films for sensitive foodstuffs, protecting the packaged goods from losing substances such as additives that are present in them or protecting them from the migration of unwanted residues in recycled intermediate layers [49].

If adjacent layers do not adhere to each other, a *tie layer* (3) can be applied to establish compatibility [50], often using PU (polyurethane) or acid/anhydride grafted polymers [19]. Alternatively, layers can be added by coating or lamination [50]. 

Better stability can be achieved by either increased thickness, using a (cheaper) filling layer, or by using a *structural layer* (4) with good mechanical properties, including tear and piercing strength [19,24]. 

Further processing requirements such as printability and print protection can be achieved by using a suitable outer *layer* (5) such as a cardboard layer or priming, or by modifying the surface of a polymer layer for good print adhesion, e.g., by using flame treatment, an electrochemical corona treatment, or fluorination [51,52]. The outer layer is often required to have a good mechanical performance [28]. Other requirements might be, for example, certain slip properties or shape stability [26], which would usually be achieved by including additives or by modifying the material preparation parameters. For the processing of the packaging materials, the inner layer and sometimes also the outer layer must often allow sealing of the contents. 

Furthermore, the print must be protected by a *coating* (6) such as a lacquer or other laminated layer. For print protection, reverse printing of the outer layer or the full composite is common [27]. 

Not all flexible packaging requires each of the above-mentioned layers. Aside from the purposes of the outer and inner layers, the intermediate layers can be randomly arranged, preferably in the way that is easiest to manufacture. While the amount, thickness, and arrangement of the packaging layers vary, some combinations of commonly used materials are frequently seen in cases where extreme requirements must be fulfilled, such as an extremely good moisture barrier for savory snacks, often provided by an aluminum layer on PP, or the usage of durable PA layers for packaging sharp-edged goods such as cheese blocks or T-bone steaks. Total packaging thicknesses usually range between 10 and 250 µm in multilayer food packaging [53]. Typical thicknesses of the individual layers are sometimes identified [26], but they are subject to frequent innovative changes. 

### 2.2. Current End-of-Life Situation for Multilayer Packaging

After the packaging use phase, the packaging will be discarded by the consumer to the German LWP collection system. If a state-of-the-art waste management system is available, the packaging will be collected, ideally in a separate fraction, transported to a sorting facility that separates the different packaging materials and polymer types, and subsequently recycled via rinsing, shredding, and regranulation. Spectroscopic methods (VNIR (visible near-infrared), NIR (near-infrared), or MIR (mid-infrared) wavelength range) in particular are used for the automated sorting of plastic waste [62,63,64]. The reflection of infrared radiation by the packaging surface discloses information about the chemical composition of the plastics, based on which the sorting decisions can be made [65,66]. In the current LWP sorting of plastic waste, the FT–NIR technique is the most commonly used [67], applying up to 50 or more single NIR sorters to separate the plastic stream into individual types of plastic (HDPE, LDPE, PET, etc.) [68]. Nested, superimposed, black, dirty, wet, or printed packaging surfaces cannot be identified properly by NIR [14]. Differentiation by color would require additional technology (VIS cameras), and identification of filling goods is not possible; therefore, further downstream recycling steps become necessary [53,69]. NIR is a surface measurement technology that penetrates only 2 µm deep into the top layer of the material [70], resulting in the detection of only the polymer layer that is facing the sensor at the time [14,15], and not identifying the other materials used for the inner layers of composites [54]. Typically, however, NIR systems can achieve sorting purities of up to 96% [71]. In industrial processes, the sorting residues amount to up to 26% of the sorting plant input [72].

As well as these identification challenges, additional impurities due to mishandling by separation air blasts frequently occur [14,70]. All this leads to undesired impurities in the secondary raw materials, and even small quantities may result in poor adhesion, lower mechanical properties, or unwanted (dark) colors in the end products, which ultimately increase the recyclers’ costs [53,73,74]. Therefore, “Der Grüne Punkt” (Cologne, Germany) usually allows impurities in the sorting fractions of about 2–8 wt% [75,76].

If a multilayer product has to be fully mechanically recycled, polymer miscibility is the key parameter. Here a distinction can be made between homogeneous (miscible), heterogeneous (immiscible), and limited miscibility systems (Table 2), with heterogeneous systems clearly predominating [77]. If the materials are compatible, i.e., fully miscible, direct regranulation is possible. The polymers PE, PP, PS, PA, and PET, which are often used in food packaging, are generally immiscible at the molecular level [78]. In the area of multilayer packaging, this heterogeneity is reinforced by the combination of other polymers and non-polymer materials [28]. Immiscible or limited miscibility multi-material structures must be separated into their layers, or at least into fractions of miscible components, which is a key waste management challenge [39,53]. 

Incompatibility in the reprocessing process is due to large differences in the specific melting temperatures or an overlap of melting and decomposition temperatures [79]. For example, the processing temperature of PET is between 270 and 300 °C, while the thermal degradation of ethylene-vinyl acetate (EVA) begins at 288 °C [79]. Immiscible polymers can be made miscible to a limited extent by compatibilizers. Such substances consist of copolymers, each with one end that is miscible and anchors itself in one of the two components of the blend. This creates strong bonds between different, incompatible polymers [77].

The better the upfront separation of the polymer waste into pure materials, the more efficient the recycling will be with conventional plastic recycling technologies [32,54]. With multilayers, this approach reaches its limits, because usually the layers cannot be separated with economically justifiable effort, and thus the low product mass along with the functionality of the packaging is achieved at the expense of recyclability [81]. Ultimately, the sum of the impurities and other contaminants in the post-consumer waste stream results in recyclates of degraded [82] or even undefinable quality [83]. Furthermore, there are currently hardly any possibilities, or only expensive test procedures, for proving the quality and composition of recyclates. This makes the use of recyclates an expensive and arduous endeavor for plastics manufacturers and recyclers, depending on who bears the costs. Furthermore, the lack of transparency and past uncertainties regarding the actual quality of plastic recyclates has led to a lack of trust between plastic recyclers and manufacturers and to insufficiently well-functioning markets for high-quality secondary raw materials [83,84]. Manufacturers are therefore increasingly turning to primary materials, as the costs are often lower and the quality fluctuates less than for secondary raw materials [83].

This highlights the need for a unified approach to recycling plastic packaging in a closed or open loop between recyclers and producers [54]. However, a corresponding system of closed material loops can only develop through the exchange of waste-related data and the cooperation of all participants along the supply chain [85]. Unlike for the recycling of mono-materials, to date there are no strategies for processing multilayer films in closed primary loops [32]. Multilayer packaging is thus emblematic of the problems that ultimately occur in waste management along the value chain of plastic packaging. The players involved focus on their interests and goals. For packaging manufacturers, for example, this leads to multilayer packaging being developed and marketed with a view to maximum functionality at minimum cost and not with a view to its recyclability. This is in contrast to the concept of CE, which aims to further develop the prevailing linear flow of plastics along the value chain by closing them into loops, so that plastic products and materials remain in the economic cycle for as long as possible [86,87]. To achieve a CE in the waste management of plastic packaging, the European Commission’s “A European Strategy for Plastics in a Circular Economy” emphasizes, among other things, recycling-friendly design and the use of innovative sorting and recycling systems [9].

Although under discussion for decades, design for recycling has so far gained little influence on the numerous suppliers in the market, and the speed of innovation for new types of packaging such as multi-material packaging does not at all match that of innovations for methods and technologies for recycling [12]. In this context, Ceflex, an initiative representing the entire value chain of flexible packaging, is trying to make all flexible packaging recyclable by 2025 through its “Designing for a Circular Economy Guidelines” [88]. The focus of phase one is on polyolefin-based flexible packaging (mono-PE, mono-PP, and PE/PP blends), as this makes up the largest part of the flexible packaging waste stream, and sorting and mechanical recycling have already been demonstrated on an industrial scale. Phase two, which is currently underway, will then address the recycling of multilayer materials. 

In addition, there are already projects working on new technological approaches that, among other things, make it possible to separate multilayer packaging from the post-consumer waste stream in a more targeted way than before. These include fluorescent marker particles, which are applied in low concentrations in or on packaging and emit a characteristic luminescence when excited in specific wavenumber ranges. The marker particles can thus contribute to improving sortability as a material-independent separation feature [89,90,91]. In the Holy Grail initiative, invisible digital water marks are applied to the surface of a package. Recycling machines can then read out the recycling information for the packaging in question [92]. 

Both approaches can be combined with the detection technologies typically used in waste management and have the potential to sort post-consumer waste streams into defined streams (e.g., food vs. non-food). More targeted sorting could enable better recycling of multilayer packaging, as it could then be treated with chemical or solvent-based delamination processes. This would pave the way to greater quantities of high-quality recyclates and enable the entire packaging value chain to take a step towards CE [93].

### 2.3. Recycling of Multilayer Materials

Each manufacturer has a choice of different material and layer combinations when developing an individual packaging solution, and this has led to an ever-increasing number of different multilayer packaging solutions (especially in the area of polymers). This prevents a clear separation into individual material groups, as is possible with packaging made of mono-materials. In the present work, the following multilayer packaging types could be identified, with the focus set on polymeric multilayers. By identifying the outer and inner layers of the packaging, a large number of sub-categories could be identified in the category *“Polymeric multilayer”* (e.g., PET–LDPE, PA–LDPE, PP–LDPE, PP–PP, PET–PET, PET–HDPE, PET–PP, LDPE–LDPE). Furthermore *“3-Composite paper/cardboard+metal+plastic”* and *“2-Composite paper/cardboard+plastic”* are mainly used for liquid cartons, *“2-Composite aluminum+paper/cardboard”* mainly consists of foils and pouches, *“2-Composite plastic+aluminum”* mainly consists of blisters and pouches, and *“Plastic+paper/cardboard unlaminated”* mainly consists of pouches, blisters, and cups. 

For many of these multilayer products, there are already several promising approaches and processes for their recycling at various degrees of maturity. These are summarized in Table 3.

With improved controllability and separability, multilayers would be of great importance for material recycling, as they represent not only a large but also a steadily growing (approximately 7% p.a. [108]) segment of the packaging market.

For a possible approximation of current multilayer shares, Equation (1) can be used. In 2010, multilayer films accounted for 17% (K0) of global film production [23]. If an annual growth rate of 7% (*p*) and a time horizon between the two studies of 9.5 years (*n*) is assumed (the LWP sorting took place in 2019), this yields a multilayer share of 32.33%. A second study assumed a multilayer share of 26 wt% [16] in the LWP stream in 2017. Taking into account a time horizon of 2.5 years, Equation (1) yields a multilayer share of 30.79%.
(1)$$K_{n}=K_{0}*{\left(1+\frac{p}{100}\right)}^{n}$$

## 3. Materials and Methods

### 3.1. Samples

The samples were taken from a Germany-wide collection of LWP covering a total of 350 participating households in 2019, as a part of the “MaReK” research project [46]. The participants were selected according to their household size and their place of residence. During a selectable two-week period from June to November 2019, they were asked not to dispose of their LWP in the usual manner but to collect it using an 80 L transparent HDPE collection bag they had been provided with, regardless of the local collection system. A total of 248 participants completed this field study and returned their collection bags via return mail to Pforzheim University. In total, 21,380 post-consumer LWP items with a total mass of 207 kg (188,869 g in this publication, as caps were not taken into account.) were analyzed, hereinafter referred to as the “original sample” [109] (see Table 4 and Table 5). Further information about the collection method in the field study was published in [46].

This research on the original sample [46] showed that identification based on the recycling code leads to an unmarked material mass share of about 31 %, or 54 % based on a count share. To analyze the packaging materials used, Fourier transform infrared attenuated total reflectance (FTIR–ATR) measurements were carried out with 1190 randomly selected packages (3467 g, caps were not considered) and additionally evaluated based on the type of packaging and the packaged goods. This representative sample, taken from the original sample, which was analyzed via FTIR-ATR, is hereinafter referred as the “IR (infrared) sample”. Table 5 shows the packaging types and their numbers within the IR sample.

Furthermore, a detailed analysis of 296 multilayer items (1828 g) was carried out. For sampling, a random selection was made from packages that were assumed to have multilayer content (see Table 6). The categories were identified through a literature review [22,27,28,110], expert interviews, and preliminary research on LWP. FTIR–ATR analysis and microscopy were used for the identification of outer and inner layers, layer thickness and number, and metallic content. This representative sample, taken from the original sample, is hereinafter referred as the “ML (multilayer) sample”.

The IR sample and the ML sample originated from the population of the original sample but were, in addition, completely independent experimental series. The analytical setup was chosen based on its simplicity, speed, and availability, in order to generate the maximum information content from the samples.

### 3.2. FTIR–ATR Analysis

For each ML sample and IR sample item, without further sample modification, both sample sides (“inner” and “outer” layer, where the “inner layer” is the layer in direct contact with the filling good) were tested. To this end, each LWP was characterized at three different, preferably unprinted, locations each on the inner and the outer layer, using an FTIR Alpha Platinum ATR (attenuated total reflectance) spectroscope from Bruker (Billerica, MA, USA) with OPUS Version 7.5 software (Bruker) and the therein contained libraries BPAD.S01, Demolib.s01, and FILLER.S01. The measurements were made in the mid-infrared range (4000–400 cm^−1^) with a resolution of 4 cm^−1^. For each spectrum, 10 scans were performed, and the arithmetic mean value was calculated. The measurements, as well as the hit quality (on a scale from 100–1000) of the assigned database spectrum, were recorded to exclude false determinations in cases of low hit qualities (<500).

### 3.3. Microscopy Analysis

From every ML sample item, a 30 mm *×* 30 mm specimen was cut out with a scalpel. The specimen was set up using specimen embedding holders and examined under a microscope. A Leica DM RM light microscope (Type 301-371.010) from Leica (Wetzlar, Germany) with LasX software (Leica) was used to provide a compiled picture. The layers were measured with the aid of the image processing tool ImageJ. The number of layers, as well as the individual thickness of each layer, was recorded.

## 4. Research Results and Discussion

### 4.1. Analysis of the IR Sample

The material distributions of the IR sample (1190 items, 3467 g) given in mass shares based on the recycling code (printed on the packaging) characterization (“labeling”) on the one hand and the IR characterization (“analysis”) on the other hand are shown in Table 7. Using the recycling code for material determination, a share of 65.71% of unmarked packaging remains. Furthermore, 10.34% of the packaging marked with “07-Other” provided no precise information on the packaging material. Thus, in total, a share of 76.05% of the packaging did not provide clear identification of its material composition. In other words, three out of four plastic packaging items in the market do not provide the customer with any information about the packaging material.

When FTIR analysis was applied, only 4.54% were not determinable, while a multilayer packaging share of 43.19% was identified, which is by far the largest single fraction of all packaging (Table 7). As expected, the second-largest fraction was composed of PP packaging (25.55%), and all other packaging materials were only found in single-digit percentage shares. Nonetheless, if the material labeling is compared with the FTIR analysis, in the cases of LDPE, PVC, and HDPE, five to six times the mass could be identified by IR analysis compared with the amount identified by the recycling code labeling information. In contrast, within the PP, PET, and PS fractions, approximately the same mass that was identifiable via labeling by a recycling code could be identified by the IR analysis. This is because PET, PS, and PP are often used for cups and trays, which often carry a recycling code.

As an intermediate result, one might state that the recycling code labeling of LWP does not work well, as only about 25% of all packaging items bear recycling code information. Moreover, most of the multilayer materials remain unlabeled, and even in the best cases of mono-material packaging (PET, PS), only about 50% of the packaging items are clearly labeled for material identification. However, if a recycling code was present on the packaging, 93% of them (PP: 95%, LDPE: 93%, PET: 93%, PS: 100%, PA: 67%, PVC: 50%, HDPE: 78%) agreed with the IR analyses performed. While the labeling policy might be relevant for consumer decisions, in industrial sorting it does not play a role as the sorting relies on material properties (NIR reflectance) but not on labels. However, appropriate labeling could lead to an improvement in LWP input quality. The need to support consumers in sorting is shown by an online survey conducted by Kantar GmbH and commissioned by the dual system in Germany in 2020, in which almost 60% of respondents stated that they needed further information for the correct separation of all types of household waste [111]. This results in a 30% share of waste that mistakenly ends up in the yellow bin (or the yellow bag) [111]. This poses a great challenge to even the most modern sorting plant and can lead to a dramatic decrease in material quality.

### 4.2. Extrapolated Original Sample, Data Validation, and Recycling Approaches

The results of the IR sample (1190 items) allowed an extrapolation to the mass of the categories “07-Other” and “Unmarked (no recycling code)” (65.5 kg or 33% mass share) of the original sample (see Table 4), assuming that both had the same composition [46]. This reduced the proportion of non-identifiable items to 2.63% (see Figure 1). As can be seen, PP (17.4%), polymeric multilayers (12.5%), PET (12.4%), 3-Composites paper/cardboard+metal+plastic (11%), and tinplate (8.5%) were the most important fractions by mass. A comparison between sorting plant output results [72] and data regarding plastic processing in the packaging industry [112] showed good consistency, with only minor differences [46]. Mono-polyolefins, at 29.3% (PP: 17.4%; LDPE: 6.95%; HDPE: 4.91%) represented the largest fraction by mass. This is in line with statements from Ceflex and also clarifies the approach of placing polyolefins at the center of the recycling of flexible packaging [88].

Furthermore, there was a total share of 33.16% of multilayers, consisting of polymeric multilayers at 12.52%, 3-Composite paper/cardboard+metal+plastic at 10.97%, 2-Composite paper/cardboard+plastic at 6.77%, and further combinations accounting for 2.90% (see “*** Other categories” within Figure 1). Measurements of the inner and outer layers of the 12.52% of polymer multilayer packaging further identified the polymers used, and these are shown as a bar chart in Figure 1. Here PET–LDPE (outer inner layer) makes up the largest share at 50.98%. This is followed by PA–LDPE at 23.29% and PP–LDPE at 4.52%. For the outer layer, PET (57.98%) and PA (24.12%) were mainly used, while the inner layer was mainly LDPE at 80.34%, followed by PP at 4.89%. For 6.66% of items, the outer, inner, or both layers could not be identified.

The 33.16% share of multilayers in the LWP stream determined in this study corresponds to the multilayer shares of 32.33% and 30.79% calculated in Equation (1). Deviations can be attributed to the authors’ definition of the term multilayer. Validation of the presented multilayer proportions could not be performed, due to a lack of studies in the literature.

Taking into account the approaches and processes for multilayer recycling presented in Table 3, there are currently, or will be in the near future (all pilot plants will be in operation in 2023), possibilities for the recycling of all 20 multilayer combinations or the 33.16 wt% identified in the present work (Figure 1). However, processes such as Purecycle or ChemPet only recycle a target polymer from the multilayer, so further post-treatment steps or processes are necessary. In addition, the Newcycling process for separating PE and PA, for example, is currently only used in a post-industrial environment. Furthermore, some processes have not yet reached large-scale industrial maturity. For example, the pilot factory of Garbo GmbH recycles 1000 t/a of PET from multilayer films and trays using the ChemPET process [105], but faces 161,800 t/a of non-recyclable PET packaging waste in Germany [72]. A second example can be given for the recycling of liquid cartons, where the plant of Saperatec GmbH can handle 18,000 t/a [98], whereas the amount of liquid cartons in Germany is over 155,600 t/a [72]. This underlines the fact that there is no lack of innovative delamination processes for the recycling of multilayer materials, but there is a lack of market maturity and industrial-scale processes. Chemical recycling processes, for example, produce large quantities of CO_2_ and are currently too expensive, due to their high energy requirements [113,114]. As a result, the accumulating quantities of multilayer packaging cannot yet be fully recycled, and the actual value for the recyclability of multilayers still has to be corrected downwards.

### 4.3. Depth Analysis of the ML Sample

The following section is intended to provide a descriptive insight into the wide variety of multilayer packaging structures. For this purpose, Figure 2, Figure 3 and Figure 4 show the number of layers, total thickness, outer and inner polymer combinations, storage conditions for packaging types, and packaged goods for the ML sample (296 multilayer items, total mass 1828 g). These parameters were chosen as they can provide considerable information to assist the recyclability of multilayer packaging. Unless otherwise stated, the percentages in chapter 4.3 are to be understood as count shares.

#### 4.3.1. Number of Layers, Total Thickness, and Outer and Inner Layer Polymer Combinations

The number of layers, the total thickness, and the outer and inner layer polymer combinations used in the multilayer packaging examined can be seen in Figure 2. Overall, the packaging examined can be divided into systems with two to eight layers, with 92% consisting of two to five layers. With a share of 38%, three-layer systems are used most frequently. The arithmetical mean (AM) number of layers for all packaging was found to be 3.7, and the AM total thickness of all packaging was 116.7 µm, with the total layer thickness of the packages ranging from 71 to 200 µm. A clear linear correlation between the AM total thickness and the AM number of layers could be established (Pearson correlation coefficients: 0.65). Observation of the materials used for the outer and inner layers shows that the outer material of 45% (count of 134) of the packaging was PET, which was thus used twice as often as PP at 22% (64) or PA at 19% (57). For the inner layers, LPDE proved to be particularly dominant, being used in 71% (209) of the packaging. PP at 21% (62) and PET at 5% (14) were other notable polymers for inner layers. Overall, the most frequently used combination was PET–LDPE at 36% (107), followed by PA–LDPE at approximately 18% (53) and PP–PP at 14.2% (42). In 24% (71) of the packaging, the same polymer was used as the outer and inner layer (PP 14% (42); LDPE 6% (17); PET 4% (11); PA 0.3% (1)).

Furthermore, there was a 20% (58) share of non-polymeric composite materials. Metallic layers such as aluminum make up the largest share at 17% (51). In some cases, there was a paper layer (0.3% (1)) or both (2% (6)). Furthermore, only the two-layer systems, representing 13% (38) of all packaging, could be fully characterized. Polymer compatibility in terms of miscibility was found in 29% (11) of the two-layer systems, due to the same polymers being used in the outer and inner layers. Limited miscibility was found in 12% (PA–LDPE (4) and PP–LDPE (1)) of the two-layer systems with different outer and inner layers. This means that in total, only 5% of the packaging examined could be recycled using conventional recycling methods. The outer or inner layers of 9% (26) of the packaging could not be identified.

In the ML sample, 13 different polymer multilayer combinations could be identified, whereas 17 were found in the IR sample. Furthermore, there were similarities concerning the most common multilayer combinations identified (see Table 8). The similarities were due to the fact that the samples were taken from the same population (original sample) and the differences were due to the different mass proportions of the ML sample (1828 g) and the IR sample (3467 g).

#### 4.3.2. Analysis of the Packaging Regarding Packaging Type and Storage Conditions

The outer and inner layer polymer combinations used, depending on the different packaging types and storage conditions, can be seen in Figure 3. The most common polymer combinations in the pouch fraction were PP–PP and PA–LDPE at 24% each (33) and PET–LDPE at 21% (28). In the film fraction, PET–LDPE at 40% (33), PA–LDPE at 23% (19), and PP–LDPE at 8% (7) were used frequently. In the stand-up pouch fraction, PET–LDPE was used most frequently at 60% (12), followed by PP–LDPE at 10% (2). In addition, within the tray fraction, PET–LDPE was used most frequently at 62% (34), followed by PP–PP at 15% (8). The highest AM total thickness of 200 µm was found within the analyzed trays, despite an AM number of layers of 3.5, which was below the AM number of layers for all packaging of 3.7. This was followed by stand-up pouches, with an AM total thickness of 119 µm, despite the highest AM number of layers of 4. The lowest AM total thickness of 94 µm was found in pouches and films. While films, with an AM number of layers of 3.4, were below the AM number of layers of all packaging, pouches had an AM number of layers of 3.8, which is above that for all packaging. Metallic components occurred in 40% (8) of the stand-up pouches and 33% (45) of the pouches. No metallic components were found in the tray fraction.

With a share of 58% (171), the majority of the ML sample packaging was stored in the refrigerator and 40% (119) at room temperature. Only 2% (6) of the packaging was stored in a deep freeze. The most frequent polymer combinations of the refrigerator fraction were PET–LDPE at 42% (71), PA–LDPE at 26% (44), and PP–PP and LDPE–LDPE at 6% each (10). In packaging stored at room temperature, PET–LDPE at 30% (36) and PP–PP at 25% (30) were mainly used. Frozen packaging consists mainly of PP–PP (33% (2)) and PP–LDPE (33% (2)). Packaging stored in the refrigerator had the highest AM total thickness of 123 µm, with the lowest AM number of layers of 3.4 of the storage options investigated. Packaging stored at room temperature had the highest AM number of layers of 4.1, but the AM total thickness of 109 µm was below that of the packaging stored in the refrigerator. Frozen packaging, despite an AM number of layers of 4, had the lowest AM total thickness of 73 µm. A total of 46% (55) of the products stored at room temperature had some metal content. In the refrigerator fraction, this was only 1% (2), while no metal content was found in frozen products.

#### 4.3.3. Analysis of the Packaging with Respect to Packaged Goods

The proportions of polymer outer and inner layers, depending on the category of packaged goods, are illustrated in Figure 4. Only packaging items of which at least ten were available were selected (see Table 6). This resulted in a total of 234 packages divided into 11 of the 25 categories. Furthermore, the category “not determinable” could contain further combinations. In 10 of the 11 categories, more than two different combinations of outer and inner layers were used to package the same goods. For mozzarella in brine, only two different material combinations (PA–LDPE and LDPE–LDPE) were found. The greatest variety of packaging consisting of different material combinations was found in sliced cheese packages (nine), followed by baked goods and sausages and cold cuts (seven each).

A comparison of food and non-food packaging shows that non-food packaging mainly used PP–LDPE (40% (6)) and PET–LDPE (33% (5)), while food packaging mostly used PET–LDPE (40% (85)) and PA–LDPE (20% (42)). Furthermore, categories that at first glance appeared to be the same could also differ. For example, PA–LDPE was used in packaging for mozzarella in brine with a share of 83% (10), while it had only a 17% (2) share for feta in brine. LDPE–LDPE was also used in packaging for mozzarella, while PET–LDPE, PP–LDPE, PA–LDPE, PP–PET, and LDPE–PA, a much wider variety of material combinations, were all used for packaging of feta in brine. Meat substitutes had the thickest packaging with an AM total thickness of 169 µm and an AM layer number of 3.6, slightly below the AM layer number for all packaging of 3.7. Packaging for sausages and cold cuts had the second-lowest AM number of layers (3.3), despite an AM total thickness of 131 µm. Only feta in brine packaging had fewer layers (3.0). The lowest AM total thickness of 71 µm was found in dry food, despite a high AM number of layers of 4. The highest AM number of layers was found in coffee (4.8) and nuts (4.6). Metallic content was found in 8 of the 11 categories shown. However, this was particularly frequent in packaging for dry food (78%), nuts (75%), and coffee, tea, and spices (70%), while no metallic materials were used in cheese packaging.

#### 4.3.4. Discussion of the ML Sample

In the investigated ML sample (296 items), systems with two to eight layers and an AM number of layers of 3.7 could be found. Furthermore, the AM total thickness of all packaging was 116.7 µm, with the total layer thickness of the packages ranging from 71 to 200 µm. The analysis of the packaging regarding the materials used showed that both PET as the outer layer and LDPE as the inner layer stood out as frequently used materials. The use of these materials was common to all the packaging types, storage conditions, and food categories investigated. The only exceptions were packaging for mozzarella and salty biscuits (crisps), as well as packaging that was stored in the freezer, for which no PET (outer layer) was used. Furthermore, LDPE (inner layer) was not used for salty biscuits. PET is used in packaging for mechanical stability, and as protection against moisture, oil/grease and aroma migration, and it also offers a good surface for printing. LDPE is suitable as an inner layer primarily because of its excellent heat sealability (low melting temperature) and inertness towards the contents. As a result, the PET–LPDE layer combination was the most commonly used across all packages at 36% (107). This combination was particularly dominant in packages for coffee, tea, and spice (60%), sausages and cold cuts (65%), nuts (50%), sliced cheese (45%), and meat substitutes (39%).

At 18% (53), the layer combination PA–LDPE was found to be the second most common. This combination is ideal for vacuum packaging of oxygen-sensitive foods such as ham, cheese, or sausages, and was therefore used in packaging for mozzarella (83%), meat substitutes (39%), sliced cheese (23%), and baked goods (23%).

The third most common material combination was PP–PP at 14% (42), with about 11% (5) of these being two-layer systems. Furthermore, PP was the only material worth mentioning besides LDPE that was used as an inner layer. The reason for the PP–PP combination is its high resistance to grease and moisture. This combination was therefore used particularly frequently in salty biscuits (79%), baked goods (39%), and dry food packaging (27%).

Aluminum was identified in 19% of the packaging within the ML sample. Aluminum protects against air, light, and moisture, and thus contributes to longer shelf life and aroma protection of the food. Therefore, aluminum was found in packaging for salty biscuits (79%), nuts (75%), and coffee, tea, and spices (70%). The same principle can be applied in reverse, as no aluminum was identified in packaging for feta in brine, mozzarella in brine, and sliced cheese, and only a 2% share in packaging was found for sausages and cold cuts. Considering the type of packaging, aluminum was detected in 40% of stand-up pouches, 30% of bags, and 1% of films, while trays contained none. In the analysis of storage conditions, it was found that 96% of the aluminum was used in food packaging stored at room temperature. While food packaging in the refrigerator contained 4% aluminum, no aluminum could be detected in packaging for frozen products. This is due to the fact that food in refrigerators and freezers is exposed to less moisture and light and requires less protection from flavor loss.

In the ML sample, only 5% (16) of the packaging examined could be recycled using conventional recycling methods. A package is defined as recyclable when all polymers can be correctly detected, and miscibility is present in the recycling process. This is only true for about 4% of the two-layer systems (13% of the ML sample) with the same outer and inner layer (PP–PP (5), LDPE–LDPE (4), and PET–PET (2)), and a proportion of about 2% (5) that despite different polymers in the outer and inner layers (PA–LDPE (4) and PP–LDPE (1)) can be recycled in a joint reprocessing process due to partial compatibility. For 9% (26) of the packaging, the outer or inner layer could not be identified, so an assessment of recyclability was not possible. This leaves a proportion of approx. 86% (254) that could not be recycled using conventional sorting methods. This is mainly due to a large number of layers with different materials that show no compatibility in the joint recycling process.

A comparison of the ML sample with the IR sample in terms of the identified polymeric multilayer combinations showed good consistency (see Table 8). Accordingly, the assessment of recyclability in Section 4.2 for the IR sample can be assumed for the proportion of conventionally unrecyclable fractions (86%) in the ML sample. This means that recycling processes or approaches for recycling the conventionally unrecyclable fractions already exist.

#### 4.3.5. Limitations of the Study

The data collected in the present work are subject to limitations due to a variety of factors influencing the LWP generated. For example, seasonal variations were not considered, due to the collection period of the LWP (June–July). Furthermore, the mass fractions of packaging shown (Table 4 and Table 7 and Figure 1) include residual content remaining in the packaging. In addition, a bias may have arisen due to the participants’ knowledge of the subsequent characterization of their LWP waste or due to the participation of those who already had a strong awareness of the disposal of their LWP waste.

## 5. Outlook

LWP waste streams in the post-consumer sector are a complex mix of different, often contaminated, material types. In particular, the multilayer packaging contained in the waste stream poses major challenges to recycling companies, due to its limited detectability, sortability, and recyclability. At the same time, actors along the value chain have contrary interests. In the present work, a share of 33.16% of the multilayer packaging, divided into 17 different multilayer packaging solutions, could be identified (see Figure 1). This variety in multilayer packaging types is problematic because each must be fed into an appropriate reprocessing process (see Table 3).

Via the presented data, insights into the mass fraction, the recyclability, and suitable recycling processes for multilayer packages can be gained. Chemical recycling processes, which are criticized for generating particularly high CO_2_ emissions, are often used. Here, the data provided can be used as input mass flows for the modeling of chemical recycling processes in the context of life cycle assessment analyses or cost calculations. In addition to the recyclability, the detection and sorting technologies represent a decisive aspect of the recyclability of packaging.

The problem of insufficient sortability in the waste management treatment of LWP still exists. Efficient sorting represents a key technology for the production of high-quality recycled polymers. The increasing digitalization of industrial processes also promises significant progress here, with further recorded information on the composition and type of waste available specifically for each item. For example, camera systems are increasingly being used in conjunction with machine learning algorithms to improve sortability. In some cases, the camera system is supplemented with other optical systems (NIR, VIS) [115]. The sorting task can include both a full sorting [116,117] of LWP streams and a sorting out of impurities [118] (silicone cartridges), the recognition of the brand, or the use of the stock-keeping unit on the packaging [117]. Insufficient sorting can, for example, lead to problems in the recycling process in the case of multilayer packaging containing aluminum, in line with the insufficient miscibility of polymers.

Flexible packaging containing aluminum is still considered by LWP waste sorters as a contaminant for recycled material and is the main cause of processing problems such as blockages of melt filters. In addition, aluminum in flexible packaging can lead to material losses during metal detection, as the metallic particles are sorted out of the line before the extruders and melt filters, in order to protect them [119]. Further, laminated and metalized aluminum leads to greying of the recyclate and is therefore not considered an optimal barrier material, but it is usually tolerated to some degree [119]. For example, AlOx coatings do not significantly affect the quality of the secondary materials, as they are typically only 1–10 nm thick and therefore often do not exceed the tolerable limit of a maximum of 5% of the total weight of the packaging structure [88]. Sorting solid metal objects, laminated films with solid aluminum layers, and laminated films with deposited aluminum layers into separate fractions would be desirable [119].

However, the creation of, e.g., object recognition systems would require a large and accurate amount of data. The additional packaging information collected in the present work, such as packaging type, filling material, or storage conditions, could contribute to a more optimal sorting of multilayer packaging, e.g., for items with aluminum content.

## 6. Conclusions

With the implementation of the European Strategy for Plastics in a Circular Economy in 2018 [9], the European Union set the course for the future achievement of a CE in the field of plastic packaging. This is urgently needed, as the recycling rate of plastic packaging in the EU is currently around 40% [87]. In the present work, the quantity and composition of multilayer packaging contained in the post-consumer waste stream was analyzed and identified as one of the problems limiting the achievement of future recycling rates, due to its limited recyclability.

Multilayer packaging enables tailor-made properties to protect a wide variety of packaged goods. Accordingly, multilayer packaging is selected based on the task to be fulfilled (protection against light, protective atmosphere, etc.), with practically no concern for its end-of-life recycling properties. The innovation speed in recycling processes does not keep up with the speed of packaging material innovations. Consequently, a lack of large-scale industrial sorting and recycling processes has led to the fact that multilayer packaging is not assigned to any specific sorting fraction but instead is dispersed into various recycling paths such as films, mixed plastics, or residual material. While residual materials, and in some cases mixed plastics, are recycled energetically, multilayer packaging represents a contaminant in the recycling of the film fraction, and in some cases also in mixed plastics, and thus it must be separated. The problem is based on the variety of polymers and non-polymeric materials used in food packaging and the differences in their specific melting temperatures or the overlap of melting and decomposition temperatures within the reprocessing process.

In overcoming these problems, the multilayer share of 33.16% in the LWP waste stream identified in this publication can not only be seen as showing the necessity for the further development of separation technologies but also as showing the potential to meet the increasing requirements of the Packaging Act (VerpackG: 63% by weight by 2022) for the recycling of plastic packaging and the potential for increasing quotas in the future.

Multilayer packaging in the area of post-consumer waste is a complex mixture of different types of materials, which are also contaminated. For the application of corresponding processes, the material composition of the multilayer packaging must be known, in order to feed it into the appropriate reprocessing process.

This problem was also recognized in this study, as IR spectrometry and microscopic examination of the outer and inner layers allowed only two-layer systems (13% of the packages in the ML sample) to undergo complete material determination.

In addition, there was also a share of 9% for which the outer or inner layer could not be identified. Furthermore, due to a multitude of different packaging solutions on the market and the sometimes too-small quantities of some multilayer packaging types, there are no economic processes for its recycling. Overall, the problem is not a lack of innovative approaches to recycling multilayer materials but rather their market maturity and industrial scale, which are not yet sufficient to recycle the current volumes of multilayers.

However, corresponding problems should not be addressed to the waste management sector alone, which, as the last link in the value chain, is often the focus of regulation and legislation. Insufficient consideration of other important factors within the areas of design, production, use, and disposal of LWP, accumulates along the value chain and makes it difficult to create a CE for plastic packaging, even with modern recycling facilities.

This underlines the fact that achieving future recycling rates, and especially a CE, will benefit from a diversity of different approaches and consideration of the entire value chain of LWP. Improvement could be provided by the introduction of innovative techniques and methods, the replacement of multilayer packaging with mono-material solutions, eco-design guidelines for distributors, the creation of a more transparent and harmonized system for all actors involved in the value chain of packaging, or legal requirements regarding standard packaging solutions per product category that would limit the possible material combinations to be managed.

To date, there are hardly any published detailed studies in the field of input analysis of LWP waste in Germany. This information gap hinders progress in waste management, as there are a large number of factors influencing the packaging waste generated, particularly in this area, which need to be investigated.

The next steps are to examine the factors influencing the packaging waste input in more detail. In particular, socio-demographic factors (e.g., household size, gender, level of education), the influence of rural or urban regions, the size of the municipality, the prevailing collection system, and the residual content remaining in the packaging could form the focus of investigations.

## Figures and Tables

**Figure 1 polymers-14-01825-f001:**
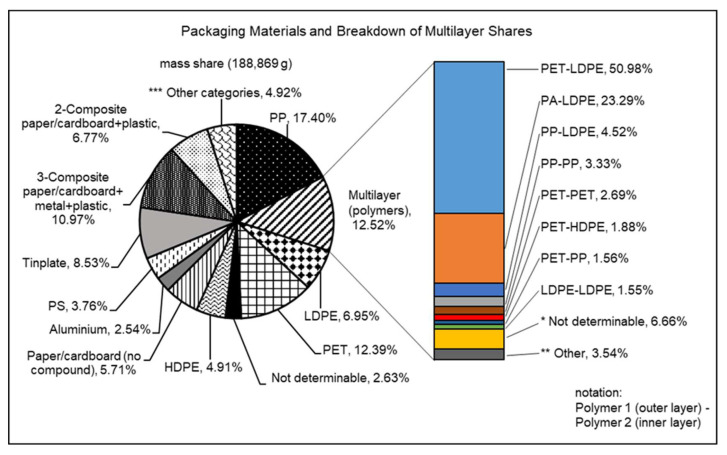
Original sample composition after IR analysis, based on [46]. * n.d.-n.d.; LDPE-n.d.; PET-n.d.; PA-n.d.; PP-n.d.; ** PP-PS; PA-PP; PA-HDPE; HDPE-HDPE; HDPE-PET; PA-PET; PA-PA; PA-PVC; PP-HDPE ***; 2-Composite aluminium+paper/cardboard, Remaining small parts, PA, Plastic+paper/cardboard unlaminated, PLA, PMMA, PVC, 2-Composite plastic+aluminium.

**Figure 2 polymers-14-01825-f002:**
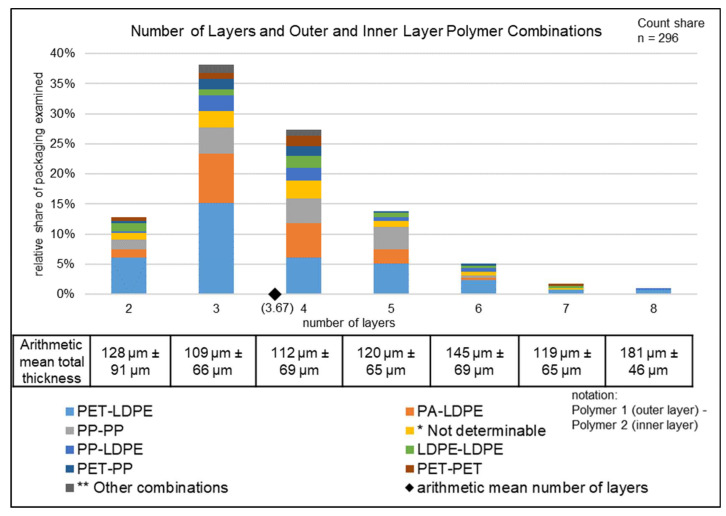
Numbers of layers and outer and inner layer polymer combinations in the ML sample. * PA-n.d.; PET-n.d.; n.d.-n.d.; n.d.-PP; n.d.-LDPE; ** PP-PET; LDPE-PET; PET-HDPE; LDPE-PA; PA-PP; PA-PA.

**Figure 3 polymers-14-01825-f003:**
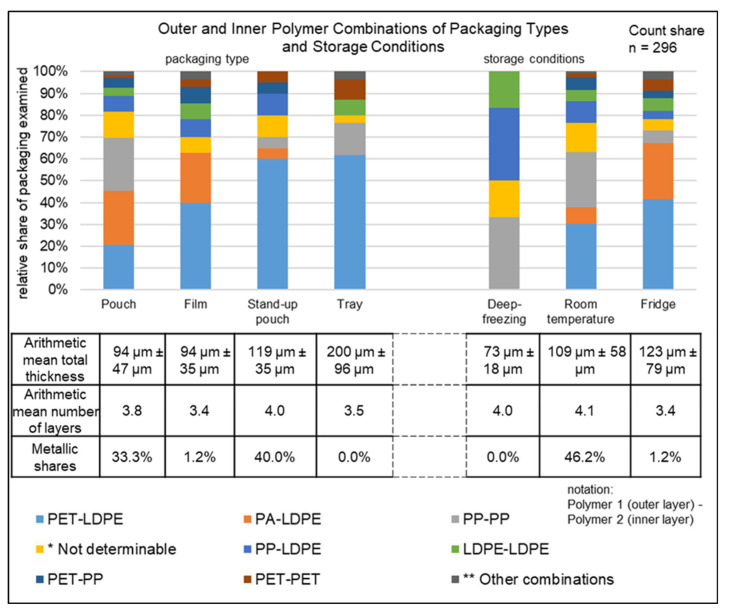
Analysis of the packaging with respect to packaging type and storage conditions. * PA-n.d.; PET-n.d.; n.d.-n.d.; n.d.-PP; n.d.-LDPE; ** PP-PET; LDPE-PET; PET-HDPE; PE-LD-PA; PA-PP; PA-PA.

**Figure 4 polymers-14-01825-f004:**
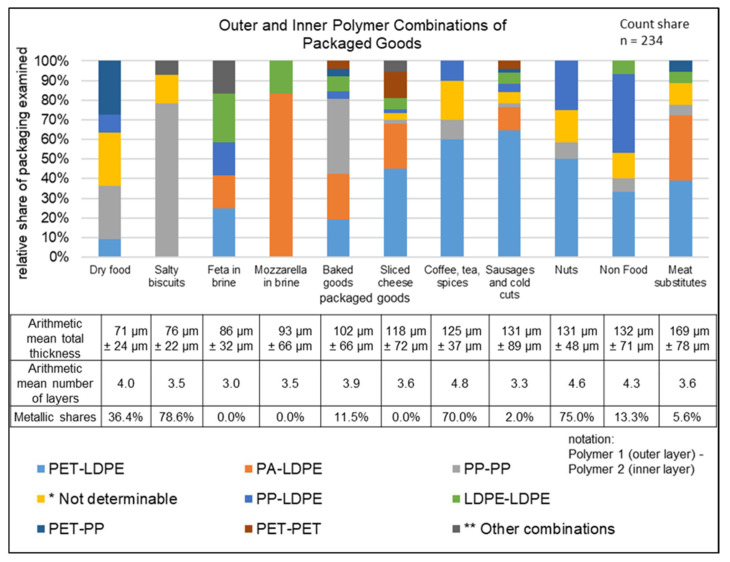
Outer and inner layer polymer combinations for different packaged goods. * PA-n.d.; PET-n.d.; n.d.-n.d.; n.d.-PP; n.d.-LDPE; ** PP-PET; LDPE-PET; PET-HDPE; HDPE-PA; PA-PA.

**Table 1 polymers-14-01825-t001:** Layers, functions, and commonly used materials for multilayer food packaging, based on [19,20,24,26,28,50,53,54,55,56,57,58,59,60,61]. Abbreviations used: LLD (linear low-density), LD (low-density), HD (high-density), PE (polyethylene), EVA (ethylene-vinyl acetate), OPP (oriented polypropylene), OPA (oriented polyamide), OPET (oriented polyethylene terephthalate), PVDC (polyvinylidene chloride), EVOH (ethylene-vinyl alcohol).

Layer	Function	Material
(1) Seal Layer (innermost layer)	Heat sealability (low melting temperature), inert against filling goods	(LLD, LD) PE, EVA, ionomers, (O)PP, (O)PA, (O)PET
	Resistance against:	
(2) Barrier Layer	Moisture	(LD, LLD, HD) PE; (O)PP, EVA, ionomers, PVDC, PET
	Oil/grease	PET, HDPE, PA, Ionomers, EVOH, PVDC
	Water vapor	PP, HDPE, PELD, PVDC
	Aroma/flavor	PET, PA, EVOH, PVDC
	Oxygen	EVOH (standard), PA or PET (below standard), Aluminum (exceeding standard), PVDC, (biaxially oriented) PA, (oriented) PET, SiOx, or Al_2_O_3_ coatings
	Light	Aluminum, TiO_2_-filled polymers
(3) Tie Layer	Combines two chemically incompatible materials	polyurethanes, acid/anhydride grafted polyolefins
(4) Structural layer	Provides shape: Toughness	PE, PET
	Puncture resistance	HDPE, PA
	Stiffness	PP, PET, HDPE, LDPE, PA, EVA, Ionomers, EVOH
	Stability	PP, PET, PA, EVA, ionomers, EVOH
(5) Outer layer	Provides printing surface and mechanical performance	PE or PET
(6) Coating (outermost layer facing environment)	Optional thin film to protect the printed material	Any specialized polymer

**Table 2 polymers-14-01825-t002:** Miscibility of different polymers, based on [80]. Abbreviations used: PS (polystyrene), PVC (polyvinyl chloride).

		Polymer-Matrix					
		PE	PP	PET	PA	PS	PVC
Additive material	PE	1	3–4	4	2–4	4	4
	PP	2–4	1	4	2–4	4	4
	PET	4	4	1	3–4	4	4
	PA	4	4	3	1	3–4	4
	PS	4	4	3	3–4	1	4
	PVC	4	4	4	4	2–4	1

Good compatibility (1); miscible up to approximately 20% (2); miscible up to approximately 5% (3); incompatible (4).

**Table 3 polymers-14-01825-t003:** Methods for the treatment of multilayers and their degrees of maturity.

Procedure/Company	Raw material/Recovery	Capacity	* TRL	Current Status
Solvent-Based Recycling Processes				
**CreaSolv**^®^ (Fraunhofer IVV) [94,95,96]	PE from post-consumer multilayer pouches	1000 t/a	7	Pilot plant (2019) for recycling post-consumer multilayer pouches in Indonesia
	PE and PP from, e.g., multilayer (post-consumer) consisting of PE/PA, PP/PET, and aluminum content	Truckload per day (approx. 5 m³ per day)	5	Construction of an industrial-scale pilot plant (2020) in Germany as part of the “Circular Packaging” project.
**Newcycling**^®^ (APK AG) [33,97,98,99]	PE/PA and aluminum from multilayer films (post-industrial) Separation of PE from PP	8000 t/a	7	Operation of a pilot plant (2018) in Germany
**Saperatec** GmbH [98,100]	PET, PE, and aluminum from each other Paper, plastic, and aluminum (liquid cartons)	18,000 t/a	5–6	Pilot plant currently under construction (completion 2023)
**Purecycle** (Procter & Gamble) [96,101]	PP from, e.g., food and liquid packaging	48,000 t/a	6	Pilot plant currently under construction (completion end 2022)
**Solvent-targeted recovery and precipitation** [102]	PE, EVOH, and PET from each other	/	1	Release (solvent) of the target polymer from the composite system with subsequent precipitation and repetition for the next target polymer.
**Recycling of post-consumer multilayer Tetra Pak^®^ packaging with the selective dissolution–precipitation process** [103]	LDPE from aluminum (Tetra Paks)	/	1	Separation through selective dissolution–precipitation process
**Chemical recycling processes**				
**ChemCycling** (BASF) [104]	Pyrolysis process enables recycling of post-consumer plastic waste (also multilayers)	/	3–4	/
**ChemPET** (Garbo) [105]	PET out of multilayer films (PET/PE/aluminum/PE) and multilayer trays (PET/PE/EVOH/PE)	1000 t/a	6	Operation of a pilot plant using glycolysis (3 t/day)
**Other approaches**				
**Recycling of multilayer packaging using a reversible crosslinking** **adhesive** [106]	PE/PET, PET/aluminum, and PE/aluminum from each other	/	1	Modification of the packaging adhesives. Separation by heated solvent from dimethylsulfoxide

* Technology Readiness Level [107]: TRL 1–3 represents proof of concept/research; TRL 4–6 represents development; and TRL 7–9 represents deployment.

**Table 4 polymers-14-01825-t004:** Packing material distribution of the original sample, based on a total of 188,869 g, based on [46].

Packaging Type	Mass Share [%]
Unmarked (no recycling code)	30.69
3-Composite paper/cardboard+metal+plastic	10.97
PP	10.46
Tinplate	8.53
PET	8.46
2-Composite paper/cardboard+plastic	6.77
Paper/cardboard (no compound)	5.71
HDPE	3.46
LDPE	3.14
PS	2.61
Aluminum	2.54
07-Other (recycling code)	2.41
2-Composite aluminum+paper/cardboard	1.57
Remaining small parts	1.27
2-Composite plastic+aluminum	0.80
Plastic+paper/cardboard unlaminated	0.53
PA	0.02
PLA	0.02
PVC	0.02
PMMA	0.01
Total	100.00

**Table 5 polymers-14-01825-t005:** Examined packaging of the IR sample, the ML sample, and the original sample.

Packaging Type	IR Sample			ML Sample			Original Sample		
	Count Share | Percentage Share [%]								
Pouch	498	|	41.85	135	|	45.61	7351	|	34.38
Foil	156	|	13.11	85	|	28.72	2466	|	11.53
Tray	301	|	25.29	56	|	18.92	2197	|	10.28
Separate closure element	3	|	0.25		|		1865	|	8.72
Cup	42	|	3.53		|		1477	|	6.91
Bag	105	|	8.82		|		1466	|	6.86
Liquid packaging	2	|	0.17		|		1027	|	4.80
Bottle	7	|	0.59		|		610	|	2.85
Can		|			|		530	|	2.48
Blister	4	|	0.34		|		494	|	2.31
Non-packaging items	7	|	0.59		|		370	|	1.73
Skin packaging	36	|	3.03		|		355	|	1.66
Tube	3	|	0.25		|		250	|	1.17
Net	6	|	0.50		|		229	|	1.07
Remaining small parts		|			|		199	|	0.93
Folding box		|			|		189	|	0.88
Other packaging element	1	|	0.08		|		113	|	0.53
Rigid foil	17	|	1.43		|		98	|	0.46
Filling material	2	|	0.17		|		60	|	0.28
Wrap packaging		|			|		25	|	0.12
Screw-top jar		|			|		9	|	0.04
* Stand-up pouch		|		20	|	6.76		|	
Total	1190	|	100.00	296	|	100.00	21,380	|	100.00

* Stand-up pouches were treated separately in the ML sample to investigate possible differences.

**Table 6 polymers-14-01825-t006:** Examined packaging within the ML sample.

Packaging for	Pouch	Foil	Tray	Stand-Up Pouch	Total
Sliced cheese	14	18	21		53
Sausages and cold cuts	5	27	20		52
Baked goods	21	4	1		26
Meat substitutes		13	5		18
Non-food items	9			4	13
Salty biscuits	13			1	14
Feta in brine		12			12
Mozzarella in brine	12				12
Nuts	7			5	12
Dry food	11				11
Coffee, tea, spices	10				10
Sweets	8				8
Preserves	2	1		4	7
Ready meals		1	3	2	6
Soft cheese	2	3	1		6
Fresh meat + fish	1	2	2		5
Granulates	5				5
Animal feed	4			1	5
Dried fruits	3			2	5
Minced meat		2	2		4
Hard cheese	1	1	1	1	4
Grated cheese	4				4
Vegetables	2				2
Butter		1			1
Rice pudding	1				1
Total	135	85	56	20	296

**Table 7 polymers-14-01825-t007:** Packaging material proportions for the IR sample (1190 items), before and after their identification by IR analysis, based on [46].

Material/Category	Result of the Recycling Code Labeling	Result after IR Analysis	Percentage Change
	Mass Share [%]		
PP	11.34	25.55	+125.30
LDPE	1.26	8.07	+540.50
PET	3.53	6.89	+95.20
PS	1.68	3.28	+95.20
PA	0.25	1.18	+372.00
PVC	0.34	2.35	+591.20
HDPE	0.76	4.96	+552.60
Multilayer	* 4.79	43.19	+801.70
Unmarked (no recycling code)	65.71	4.54 remain	−93.10
07-Other	10.34	not determinable	
Total	100.00	100.00	

* More recycling codes on the packaging or clear allocation to multilayer fraction (e.g., butcher film (plastic + paper)).

**Table 8 polymers-14-01825-t008:** Polymer multilayer combinations of the ML sample and IR sample in wt%.

Material Combinations	PET–LDPE	PA–LDPE	PP–PP	PP–LDPE	LDPE–LDPE	PET–PET	PET–PP
ML sample (wt%)	38	15	14	8	6	5	3
IR sample (wt%)	51	23	3	5	2	3	2

## Data Availability

The data presented in this paper are used in the context of ongoing research projects. More in-depth data cannot yet be provided.

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
