# Peer review of "Multilayer Packaging in a Circular Economy"

_polymers, 2022, doi:10.3390/polym14091825_

Round 1

Reviewer 1 Report

This manuscript is near impossible to review. The manuscript contains eight table, but only one table (Table 3) is referred to in the text.

Figure 2 is hard to read. Maybe replace the patterns with colours.

The terms "IR sample" and "ML" sample are not clearly explained. I understand what the abbreviation stands for, but I struggle to understand which samples they represent. What are samples "ML13" or "IR 17"? Why select just those two? How are those two samples the typical representatives of those types?

Please explain why there is a very large proportion of samples that could not be determined.

Reviewer 2 Report

The introduction should be reframed. Relevant gaps from the literature should be clarified. The context should be briefly introduced (circular economy is cited only at the end without any reference). Also the importance of the digital technologies should be highlighted and related to the CE principles. A reminder should be added at the end. 

Section 2: this section should be improved. Literature is poor. CE and technologies should be introduced better.

Please, look at the main systematic literature reviews on these domains:

Acerbi, F. et al. (2021) ‘A Systematic Literature Review on Data and Information Required for Circular Manufacturing Strategies Adoption’, Sustainability, 13(2047), pp. 1–27. doi: https://doi.org/10.3390/su13042047.

Rosa, P. et al. (2020) ‘Assessing relations between Circular Economy and Industry 4.0: a systematic literature review’, International Journal of Production Research, 58(6), pp. 1662–1687. doi: 10.1080/00207543.2019.1680896.

Sassanelli, C., Rosa, P. and Terzi, S. (2021) ‘Supporting disassembly processes through simulation tools: A systematic literature review with a focus on printed circuit boards’, Journal of Manufacturing Systems, 60, pp. 429–448. doi: 10.1016/J.JMSY.2021.07.009.

The research context should be introduced first in a more general way and then argued in relation with the specific industry (multilayer packaging).

Please revise references to the tables and figures:

"Each layer adds an important function to the overall architecture. The in-85 tended functions and the layers used to achieve them, as well as the materials used to 86 fulfill these functions, are listed in Error! Reference source not found.."

It is not clear how the research methodology has been built.

Section 5 (outlook or better "discussion") is poor. A more in depth discussion of the results should be provided and contributions to knowledge and practice and managerial implications clarified.

Section 6: limitations of the study should be better argued together with future researches.

Round 2

Reviewer 1 Report

The authors of the manuscript "Multilayer packaging in a circular economy" have taken into account the comments from both reviewers in a satisfactory manner. The manuscript is much easier to read and understand following the changes.

I have however, one more comment that I missed in my initial review. In Table 7, the authors report the packaging material according to the recycling codes indicated on the labels and the packaging material according to their own IR analyses. That IR information shows the identity of much of the unmarked packaging material. However, it would make a lot of sense if the authors could indicate whether their IR analyses agreed with the recycling codes on the labels. For instance, according to the labelling code 11.34% of material was PP; 1.26% was LDPE, etc. Can the authors confirm that the labelling codes were correct after they carried out their own analyses? The authors can either make a statement in the text or add an additional column to their table 7.

Reviewer 2 Report

The authors addressed all the issues raised.

I suggest some contributions to consider when dealing with CE and digital technologies:

[1]      C. Sassanelli, P. Rosa, and S. Terzi, “Supporting disassembly processes through simulation tools: A systematic literature review with a focus on printed circuit boards,” J. Manuf. Syst., vol. 60, pp. 429–448, Jul. 2021, doi: 10.1016/J.JMSY.2021.07.009.

[2]      R. Rocca, P. Rosa, C. Sassanelli, L. Fumagalli, and S. Terzi, “Integrating Virtual Reality and Digital Twin in Circular Economy Practices: A Laboratory Application Case,” Sustain., vol. 12, no. 2286, 2020.

[3]      C. J. Chiappetta Jabbour, P. D. C. Fiorini, N. O. Ndubisi, M. M. Queiroz, and É. L. Piato, “Digitally-enabled sustainable supply chains in the 21st century: A review and a research agenda,” Sci. Total Environ., vol. 725, p. 138177, Jul. 2020, doi: 10.1016/j.scitotenv.2020.138177.

[4]      S. Gupta, H. Chen, B. T. Hazen, S. Kaur, and E. D. R. Santibañez Gonzalez, “Circular economy and big data analytics: A stakeholder perspective,” Technol. Forecast. Soc. Chang., vol. 144, pp. 466–474, 2019, doi: 10.1016/j.techfore.2018.06.030.

These are only some of the contributions dealing with CE linked with digital technologies. Further literature could be analysed.
